# The impact of different antimicrobial exposures on the gut microbiome in the ARMORD observational study

**Leon Peto[1,2]\*, Nicola Fawcett[1,2], Musaiwale M Kamfose[1], Claire Scarborough[1], Andy Peniket[1], Robert Danby[1,3], Timothy EA Peto[1,2,4], Derrick W Crook[1,2,4], Martin J Llewelyn[5,6†], Ann Sarah Walker[1,2,4†]**

[1]Oxford University Hospitals NHS Foundation Trust, Oxford, United Kingdom; [2]Nuffield Department of Medicine, University of Oxford, Oxford, United Kingdom; [3]Anthony Nolan Research Institute, Royal Free Hospital, Hampstead, London, United Kingdom; [4]NIHR Health Protection Research Unit in Healthcare Associated Infections and Antimicrobial Resistance, University of Oxford, Oxford, United Kingdom; [5]Brighton and Sussex Medical School, Falmer, United Kingdom; [6]University Hospitals Sussex NHS Foundation Trust, Brighton, United Kingdom

**\*For correspondence:**
leon.peto@ndm.ox.ac.uk

†These authors contributed equally to this work

## eLife Assessment

This study offers a **valuable** assessment of the impact of antibiotics on the human gut microbiota across diverse observational cohorts. The findings presented are **convincing**, despite the observational design and residual confounding that may still contribute to discrepancies between the cross-sectional and longitudinal data. The work is relevant for researchers and clinicians interested in antimicrobial resistance and the impact of antibiotics on the host.

**Abstract** Better metrics to compare the impact of different antimicrobials on the gut microbiome would aid efforts to control antimicrobial resistance (AMR). The Antibiotic Resistance in the Microbiome – Oxford (ARMORD) study recruited inpatients, outpatients, and healthy volunteers in Oxfordshire, UK, who provided stool samples for metagenomic sequencing. Data on previous antimicrobial use and potential confounders were recorded. Exposures to each antimicrobial were considered as factors in a multivariable linear regression, also adjusted for demographics, with separate analyses for those contributing samples cross-sectionally or longitudinally. Outcomes were Shannon diversity and relative abundance of specific bacterial taxa (*Enterobacteriaceae*, *Enterococcus*, and major anaerobic groups) and antimicrobial resistance genes (targeting beta-lactams, tetracyclines, aminoglycosides, macrolides, and glycopeptides). 225 adults were included in the cross-sectional analysis, and a subset of 79 patients undergoing haematopoietic cell transplant provided serial samples for longitudinal analysis. Results were largely consistent between the two sampling frames. Recent use of piperacillin-tazobactam, meropenem, intravenous co-amoxiclav, and clindamycin was associated with large reductions in microbiome diversity and reduced abundance of anaerobes. Exposure to piperacillin-tazobactam and meropenem was associated with a decreased abundance of *Enterobacteriaceae* and an increased abundance of *Enterococcus* and major AMR genes, but there was no evidence that these antibiotics had a greater impact on microbiome diversity than iv co-amoxiclav or oral clindamycin. In contrast, co-trimoxazole, doxycycline, antifungals, and antivirals had less impact on microbiome diversity and selection of AMR genes. Simultaneous estimation of the impact of over 20 antimicrobials on the gut microbiome and AMR gene abundance highlighted important differences between individual drugs. Some drugs in the WHO Access group

(co-amoxiclav, clindamycin) had similar magnitude impact on microbiome diversity to those in the Watch group (meropenem, piperacillin-tazobactam) with potential implications for acquisition of resistant organisms. Metagenomic sequencing can be used to compare the impact of different antimicrobial agents and treatment strategies on the commensal flora.

## Introduction

Effective antimicrobial stewardship is necessary to limit the emergence and spread of antimicrobial resistance (AMR; *O'Neill, 2016*), and this includes the preferential use of agents least likely to select for drug-resistant pathogens among the commensal flora. This typically consists of avoiding 'broad' spectrum antimicrobials, as well as those to which resistance is currently rare, and these principles underlie the World Health Organisation AWaRe classification (Access, Watch, and Reserve; *Sharland et al., 2022*). AWaRe is reflected in the current UK National Health Service standard contract, which requires hospitals to increase the proportion of 'Access' antibiotics used, replacing previous requirements to reduce overall antibiotic use (*NHS England, 2022*). However, these classifications are largely based on activity against pathogens rather than direct measures of AMR gene selection or microbiome disruption, and stewardship could be improved if such measures were available. For example, two antibiotics being considered for use may have similar spectra against a target pathogen, but very different impacts on AMR selection, either because the amounts reaching the commensal flora differ, or because they have differing spectra against non-pathogenic commensals that protect against colonisation with drug-resistant pathogens. Metagenomic sequencing provides a direct measure of AMR genes and microbiome composition in a sample, and its increasing availability in recent years is starting to allow the fuller impacts of antimicrobials to be measured.

The large intestine contains the vast majority of human commensal bacteria and is the primary reservoir for several clinically important commensal pathogens, particularly the *Enterobacteriaceae* (including *Escherichia coli* and *Klebsiella pneumoniae*) and *Enterococcus faecium*. Increasing multidrug resistance in these organisms represents a major global public health challenge (*Ruppé et al., 2015*; *Miller et al., 2014*). Existing evidence linking antibiotic exposure to individual-level selection for AMR in the gut flora comes predominantly from small, healthy volunteer studies, which have shown that antibiotics can cause rapid microbiome disruption but provide limited comparative data between antibiotics. They may also have limited applicability to real patients, who often have recent exposure to several different agents and are at high risk of colonisation with drug-resistant organisms. Randomised trials are a robust method to assess the impacts of different treatment approaches, but few have reported microbiome outcomes (*Edlund et al., 2022*; *Reyman et al., 2022*; *Vehreschild et al., 2022*; *Pickering et al., 2022*). Another approach is to exploit variation in routine use of antibiotics in groups of patients at high risk of AMR to understand the nature and extent of differences between agents which cannot easily be achieved in other designs. Here, we report results from a prospective observational study assessing the impact of antimicrobial use on the gut microbiome and selection of AMR genes. This study included two different sampling frames to produce independent and complementary estimates, one cross-sectional, analysing a single stool sample from each participant, and one longitudinal, analysing changes in serial samples taken from the same participant admitted for haematopoietic cell transplant to enrich for broad-spectrum antimicrobial exposure.

## Materials and methods
### Study design and participants

The Antibiotic Resistance in the Microbiome – Oxford (ARMORD) study was an observational study that recruited healthy individuals living in Oxfordshire, and patients at the Oxford University Hospitals NHS Foundation Trust (OUH). The study was coordinated by the Nuffield Department of Medicine, University of Oxford, and was approved by the East Midlands-Leicester Central Research Ethics Committee (15/EM/0270).

The study involved two sampling strategies:

1. *Cross-sectional sampling.* Participants provided a single stool sample, and measures of the microbiome and AMR gene abundance were related to exposures recorded at the time of sampling.

2. *Longitudinal sampling.* Participants provided serial stool samples, and changes in the microbiome and AMR gene abundance between serial samples from the same individual were related to exposures between samples. Longitudinal sampling was only performed in patients admitted to the OUH haematology ward for haematopoietic cell transplant (HCT). The initial sample collected from these patients was also used in the cross-sectional analysis.

Participants were eligible if they were ≥18 years old, had no stoma or active inflammatory bowel disease, and were able to give informed consent and provide a history of recent antibiotic use. General medical inpatients and outpatients at OUH were approached by a member of the study team during routine care (without regard to the reason for admission or attendance), and healthy individuals responded to articles in local media. After providing written informed consent, participants were interviewed and their medical notes were reviewed to collect information about antimicrobial exposures in the past year, recent travel, diet, alcohol and tobacco use, animal exposures, healthcare exposure and use of specific drugs (case report form in *Supplementary file 1*). Electronic patient records available at OUH included (i) inpatient, emergency department, and outpatient attendances; (ii) inpatient and discharge antimicrobial prescriptions; (iii) microbiology, haematology, and biochemistry results; (iv) inpatient clinical observations (used to calculate the National Early Warning Score 2 [NEWS2] summary score of physiological abnormality; *Royal College of Physicians, 2017*); and (v) discharge coding (including Charlson comorbidity index).

## Sample collection

In the cross-sectional stratum, participants were asked to provide the first stool sample passed after recruitment. This was stored at ambient temperature for a maximum of 24 hr before being frozen at –80 °C. In the longitudinal stratum, participants were asked to provide a stool sample every other day until discharge. These were stored at 4–8°C for up to 72 hr before being frozen at –80 °C.

## DNA extraction, sequencing, and bioinformatic analysis

DNA extraction was performed by bead beating in Lysing Matrix E tubes (MP Biomedicals) followed by QIAGEN Fast DNA Stool MiniKit (QIAGEN; details in Appendix p1). Samples were sequenced in batches of 56–114 at the Oxford Centre for Genomics and had automated normalisation and library preparation using either NEBNext Ultra or NEBNext Ultra II FS kits. All samples from the same batch were pooled and sequenced across 2–8 Illumina HiSeq4000 lanes using 150 bp paired-end sequencing. Following human read removal, all samples were subsampled to a depth of 3.5 million paired reads, and samples with fewer reads were excluded. Taxonomic classification was performed with MetaPhlAn2 (for diversity indices) and Kraken2 (for abundance of specific taxa). AMR gene detection in metagenomic sequence data was performed with the ARIBA software package, using the CARD database and ontology (v3.0.2; details in Appendix p2).

## Outcomes

Sequence data was used to derive three types of outcome (further details in Appendix p3):

1. *Shannon diversity index* - a single metric of diversity for each sample (calculated as the sum of -p*ln(p) for all species, where p is proportional abundance). This index incorporates relative abundance and the number of species detected (richness).
2. *Log relative abundance of specific bacterial taxa.* The taxa of interest were two major groups of opportunistic pathogens (family *Enterobacteriaceae* and genus *Enterococcus*), and three major groups of anaerobes that make up the majority of the gut microbiome in most people but are largely non-pathogenic, so may be important for colonisation resistance (phyla Bacteroidetes and Actinobacteria, and class Clostridia). Relative abundance was the proportion of reads in the sample that mapped to a group. If a particular taxon was not detected, its relative abundance was imputed as $10^{-6}$ (i.e. a pseudocount at the approximate lower limit of detection).
3. *Log relative abundance of specific classes of AMR genes.* Five classes of clinically important resistance mechanisms were of interest: Clinically significant beta-lactamases (CTX-M, OXA, TEM, SHV), tetracycline ribosomal protection proteins, aminoglycoside transferases (AAC, ANT, APH), macrolide/clindamycin resistance genes (erm and mef), and the vanA vancomycin resistance gene. If a particular gene was not detected, its relative abundance was imputed as $10^{-5}$ (i.e. a pseudocount at the approximate lower limit of detection).

## Statistical analysis

A separate model was fitted for each of the eleven outcomes (Shannon diversity, five bacterial taxa, and five classes of AMR genes, as above). Model results are presented side-by-side for the five bacterial taxa and for the five AMR gene classes. For bacterial taxa and AMR genes, the outcomes are relative (not absolute) abundance, so the effects of antimicrobials on different taxa/genes are not independent (e.g. if the only effect of an antimicrobial was to eradicate one major taxon and leave the absolute abundances of others unchanged, then the relative abundances of these other taxa would increase). Analysis was performed in R v4.2.3.

## Cross-sectional sampling frame

Multivariable linear regression was used to estimate the effects of specific antimicrobial exposures on the outcomes above. Covariates were: age, sex, participant category (healthy, general medical, autologous stem cell transplant, allogeneic stem cell transplant), days of chemotherapy received (0 for non-HCT participants, truncated at 14), maximum Charlson comorbidity score in the year before sampling (identified from electronic health records), and the following physiological abnormalities in the fortnight prior to sample collection; maximum NEWS2 score, C-reactive protein (CRP) >50 mg/L, white cell count (WCC) >11 × $10^9$/L, and WCC <0.5 × $10^9$/L. Healthy volunteers lived in the OUH catchment area but most had no previous activity at OUH, and in this case normal values were imputed (i.e. Charlson Index 0, NEWS2 score 0, CRP <50 mg/L, WCC <11 and>0.5 × $10^9$/L). Other covariates such as diet and travel were not included in the final model, as they were not significantly associated with Shannon diversity in multivariable analysis and their inclusion did not materially affect other estimates. We did not make a formal adjustment for multiple testing and present unadjusted p-values, which should be interpreted in the light of the number of outcomes (11) and antimicrobials (21) considered. Many of the antibiotic exposures were correlated or reflected iv/oral administrations of antimicrobials from the same class, and we considered relative abundances as described above, meaning that it is not straightforward to correct for multiple testing without being overly conservative. We therefore chose to interpret the findings as exploratory in the context of the supporting level of evidence, the number of comparisons performed and consistency across outcomes.

All antimicrobial exposures observed in >5 participants were included in the model. Individual agents were categorised separately if given by different administration routes (e.g. oral vancomycin and intravenous vancomycin), but route of administration was ignored in analyses of antimicrobial class (e.g. glycopeptides). In categorising antimicrobial class, beta-lactams were divided into 'narrow spectrum' (defined as penicillin, amoxicillin, flucloxacillin and first-generation cephalosporins) and 'broad-spectrum' (all others). Exposure to each antimicrobial was included as a separate variable on a scale of 0 (no exposure) to 1 (full exposure). In order to reflect both recency and total duration of antimicrobial use, this exposure was modelled as the area under an exponential decay curve of the form y=2$^{x/\lambda}$, in which $\lambda$ is the microbiome disruption half-life, and x is time before sample collection. A single value of $\lambda$ was used for all analyses, chosen as the common value across all antimicrobial exposures with the lowest Akaike Information Criterion across 1–14 days in the cross-sectional model for microbial diversity (6 days; *Appendix 1—table 1*). The disruption half-life of 6 days means that after 6 days of an antibiotic course, a patient would have an exposure of 0.5 to that agent, and after 12 days they would have an exposure of 0.75. Details of the exposure calculation, including graphical depictions, are in the Appendix (p3 and *Appendix 1—figures 1 and 2*).

## Longitudinal sampling frame

Serial samples collected from participants undergoing HCT were used for the longitudinal analysis. The unit of analysis was a pair of consecutive samples collected from the same individual. For patients with >2 samples, each consecutive pair was used (i.e. sample 2 in pair 1 was sample 1 in pair 2, and so on, so the total number of pairs per participant is one less than the number of samples). Only pairs of samples collected within 2–30 days of each other were used. A multivariable linear regression model was used that was analogous to the cross-sectional model described above, except that the outcome was the *change* between the first and second samples in a pair, rather than absolute values (i.e. change in Shannon diversity, or change in log relative abundance of bacterial taxa or AMR genes). Because pairs of samples from the same participant may not be independent of one another, robust standard errors were used to allow for possible clustering.

**Table 1.** Characteristics of participants in cross-sectional analysis.

| | Healthy volunteers (n=33) | General medical patients (n=91) | HCT patients (n=101) | All participants (n=225) |
|---|---|---|---|---|
| Age, years (median, IQR) | 37 (31–49) | 76 (67–83) | 58 (50-66) | 64 (50–73) |
| Sex (n, %) | | | | |
| Male | 7 (21%) | 53 (58%) | 60 (59%) | 120 (53%) |
| Female | 26 (79%) | 38 (42%) | 41 (41%) | 105 (47%) |
| Recent antibiotic use (n, %) | | | | |
| Receiving antibiotics at time of sampling | 0 (0%) | 55 (60%) | 42 (42%) | 97 (43%) |
| Use in past month (but not at time of sampling) | 3 (9%) | 26 (29%) | 22 (22%) | 51 (24%) |
| Use in past year (but not in past month) | 4 (12%) | 6 (7%) | 33 (33%) | 43 (19%) |
| No antibiotics in past year | 26 (79%) | 4 (4%) | 4 (4%) | 34 (15%) |
| Max Charlson index in past year (median, IQR)* | 0 (0–0) | 4 (0–13) | 0 (0–8) | 0 (0–8) |
| Maximum values in past 14 days (median, IQR) | | | | |
| NEWS2* | 0 (0–0) | 5 (2-8) | 3 (2-4) | 3 (1-5) |
| C-reactive protein† | 0.2 (0.2–0.2) | 63 (22–163) | 10 (3–69) | 21 (2–81) |
| White cell count‡ | 7.5 (7.5–7.5) | 11.5 (8.4–14.4) | 7.6 (5.8–10.9) | 8.4 (7.4–12.4) |
| Days of chemotherapy at time of sampling (median, IQR) | 0 (0–0) | 0 (0–0) | 3.0 (1.4–7.2) | 0 (0–2.8) |

*Imputed as 0 if no observations recorded, see Methods. NEWS2 is National Early Warning Score 2.

†Imputed as 0.2 if no result recorded.

‡Imputed as 7.5 if no result recorded.

Covariates were: age, sex, type of transplant (allogeneic or autologous), days of chemotherapy received at collection of sample 1 (truncated at 14), number of days between sample 1 and sample 2, change in NEWS2 score between sample 1 and sample 2, and the presence of the following new physiological abnormalities recorded after collection of sample 1 and before collection of sample 2; CRP >50 mg/L, WCC >11 × $10^9$/L, WCC <0.5 × $10^9$/L. For each outcome, the value for the first sample in the pair was also included as a covariate (i.e. baseline diversity or log relative abundance of taxa/ AMR genes).

Antimicrobial exposures were calculated for samples as in the cross-sectional analysis above, and the exposure for each pair was the difference between the first and second samples. This has the following implications: (i) if a patient starts an antimicrobial after the first sample is collected, then the exposure for that pair is the same as for sample 2, (ii) if a patient is on long-term antimicrobial treatment, then the exposure for a pair is zero (as one would not expect this to lead to a difference between samples), and (iii) if a patient stops antimicrobial treatment shortly after sample 1, then the exposure to that agent will be negative (as one expects gut microbiome diversity to increase after stopping an antimicrobial). Truncating the small number of negative values at zero had little impact on results (data not shown).

## Results

Between July 2015 and November 2018, 225 participants were recruited and had at least one sequenced stool sample, all of whom were included in the cross-sectional analysis (*Table 1* and *Appendix 1—figure 3* [CONSORT diagram]). Thirty-three (15%) were healthy volunteers, 91 (40%) were general medical patients (84 acute admissions and 7 attending general outpatient clinics), and 101 (45%) were HCT patients. The healthy volunteers were, on average, younger, more likely to be female, and rarely had recent antibiotic exposure. In contrast, 184/192 (96%) of medical and haematology patients had received antibiotics in the past year, and 97 (51%) of these were receiving antibiotics at the time of

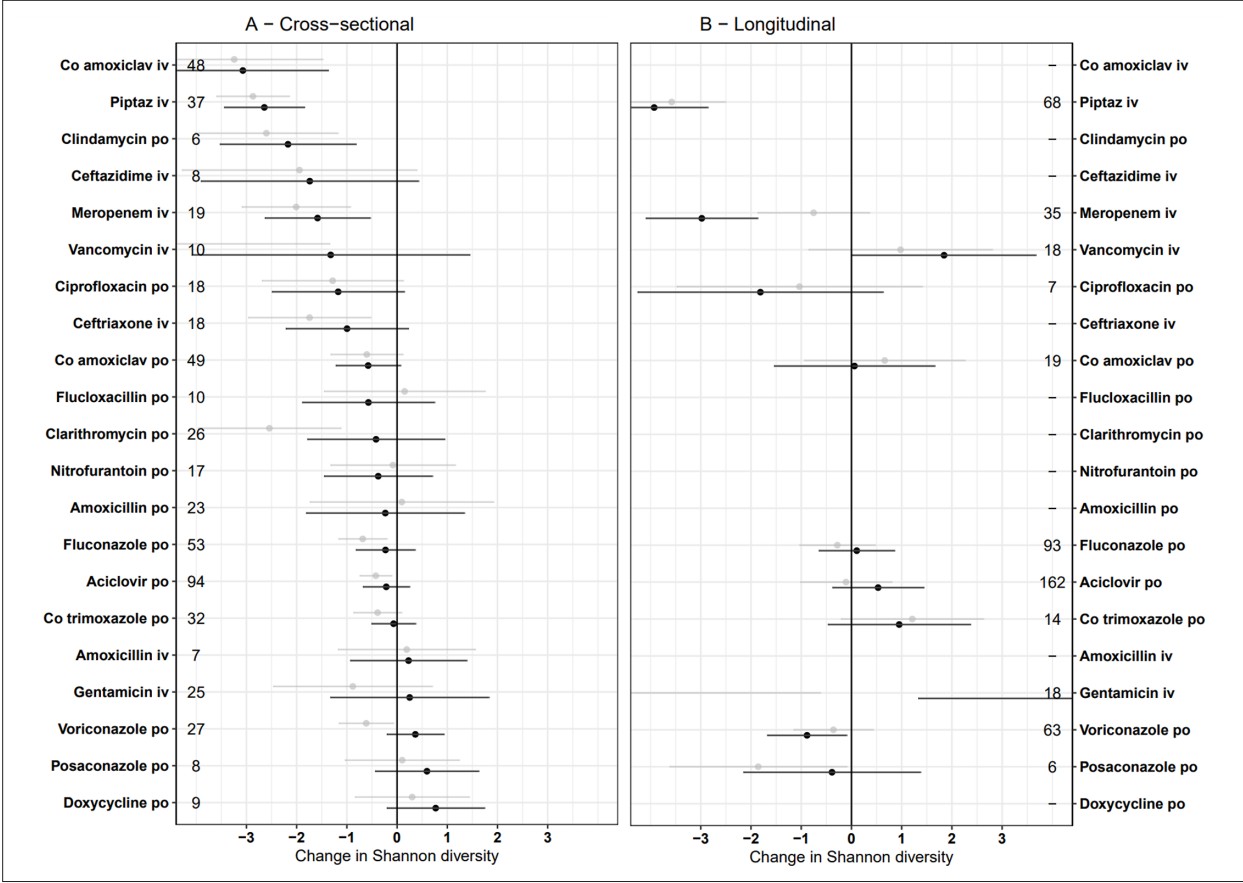

**Figure 1.** Independent effect of specific antimicrobial exposures on Shannon diversity in (A) cross-sectional and (B) longitudinal analyses. Multivariable estimates are in black, univariable (unadjusted) estimates in grey. Error bars represent 95% confidence intervals. Numbers by exposure represent sample size (n). Non-antimicrobial covariates are not shown but were included in the model and can be found in *Source data 1*. Results are not plotted for four antimicrobials (n=6–12) that had standard errors >3 and did not differ significantly from zero. Estimates represent the impact of prolonged use, when exposure ≈ 1 (approximately 42 days, see *Appendix 1—figure 2*).

sampling. Of those with antibiotic exposure in the past month, 99/148 (67%) had received >1 type of antimicrobial. Of 101 HCT participants in the cross-sectional analysis, 79 had >1 sequenced sample and so also contributed to the longitudinal analysis, and 173 sample pairs were included in this analysis. Bacteroidetes and Clostridia were predominant in most sequenced samples, and along with Actinobacteria, *Enterobacteriaceae*, and *Enterococcus*, accounted for a median 91% of classified organisms in baseline samples (*Appendix 1—table 2*). All these taxa were detected in all samples, apart from three samples with no detectable *Enterobacteriaceae*. The clinically significant AMR mechanisms included in this analysis were frequently detected: beta-lactamases in baseline samples from 67 (30%) patients, tetracycline AMR genes in 214 (95%), aminoglycoside transferases in 166 (74%), macrolide/clindamycin AMR genes in 210 (93%), and vanA in 31 (14%). Microbiome and resistome profiles of all samples are included in the *Source data 1*.

## Impact of antimicrobial exposures on gut microbiome diversity

A half-life of 6 days was identified as the best fit for the antimicrobial exposure in the cross-sectional model of microbiome diversity, and this value was used for all subsequent analyses (*Appendix 1—table 1*). The independent effects of antimicrobial exposure on gut microbiome diversity in the cross-sectional and longitudinal models are shown in *Figure 1*. The cross-sectional model (*Figure 1A*) provided more precise estimates of microbiome disruption than the longitudinal model, and four antimicrobial exposures were associated with a significant reduction in gut Shannon diversity: iv co-amoxiclav (–3.0 95% Confidence Interval –4.7 to –1.4; p=0.0005), iv piperacillin-tazobactam (–2.6, [-3.4 to -1.8]; p<10⁻⁹), oral clindamycin (–2.2 [-3.5 to -0.8]; p=0.002), and iv meropenem (–1.6 [-2.6 to -0.5];

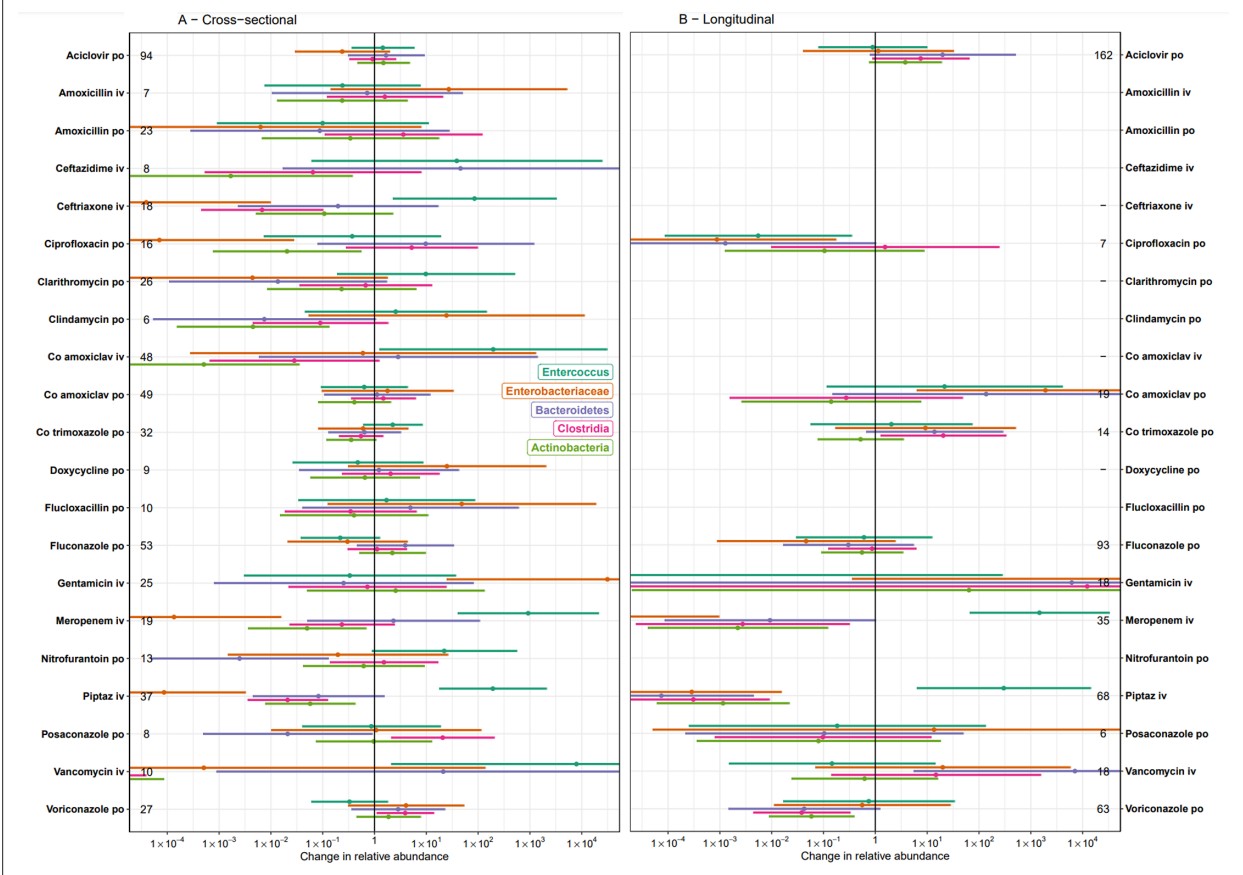

**Figure 2.** Independent effects of specific antimicrobial exposures on relative abundance of selected taxa, in (A) cross-sectional and (B) longitudinal multivariable analyses. Error bars represent 95% confidence intervals. Numbers by each exposure represent sample size (n). Non-antimicrobial covariates are not shown but were included in the model and can be found in *Source data 1*. Estimates represent the impact of prolonged use, when exposure ≈ 1 (approximately 42 days, see *Appendix 1—figure 2*).

p=0.003). There was no evidence that the impact of iv co-amoxiclav was greater than iv meropenem (p=0.13). In contrast, co-trimoxazole and doxycycline were associated with minimal reductions in gut microbiome diversity, as were azole antifungals and acyclovir (lower 95% CI above –1.0). In the longitudinal analysis (*Figure 1B*), only five exposures had data from more than 20 sample pairs, and these were consistent with the cross-sectional results, including large reductions in diversity associated with exposure to piperacillin-tazobactam (–3.9 [-5.0 to -2.9]; p $<10^{-10}$) and meropenem (–3.0 [-4.1 to -1.9]; p $<10^{-6}$; data on iv co-amoxiclav and oral clindamycin insufficient for comparison). In the longitudinal model, gentamicin was associated with marginally increased gut microbiome diversity (+6.6 [+1.3 to+11.8]; p=0.02).

Analyses by antimicrobial class were also largely consistent between cross-sectional and longitudinal analyses (*Appendix 1—figure 4*). In both analyses, narrow-spectrum beta-lactams were associated with substantially lower microbiome disruption than broad-spectrum beta-lactams. In the cross-sectional analysis, several other classes of antibiotics also had little to no impact on the microbiome diversity; antifolates, macrolides, and tetracyclines (lower 95% CI above –1.0).

## Effects of antimicrobials on the abundance of specific taxa and AMR genes

Piperacillin-tazobactam, meropenem, and ciprofloxacin were associated with significant decreases in the relative abundance of *Enterobacteriaceae* in both models, as were ceftriaxone and ceftazidime in the cross-sectional model (>1000 fold decreased relative abundance, p<0.01, *Figure 2*, *Appendix 1—figure 5*). There was no evidence of effects of iv co-amoxiclav (p=0.90) or oral clindamycin (p=0.31) on relative abundance of *Enterobacteriaceae*. Piperacillin-tazobactam and meropenem were also

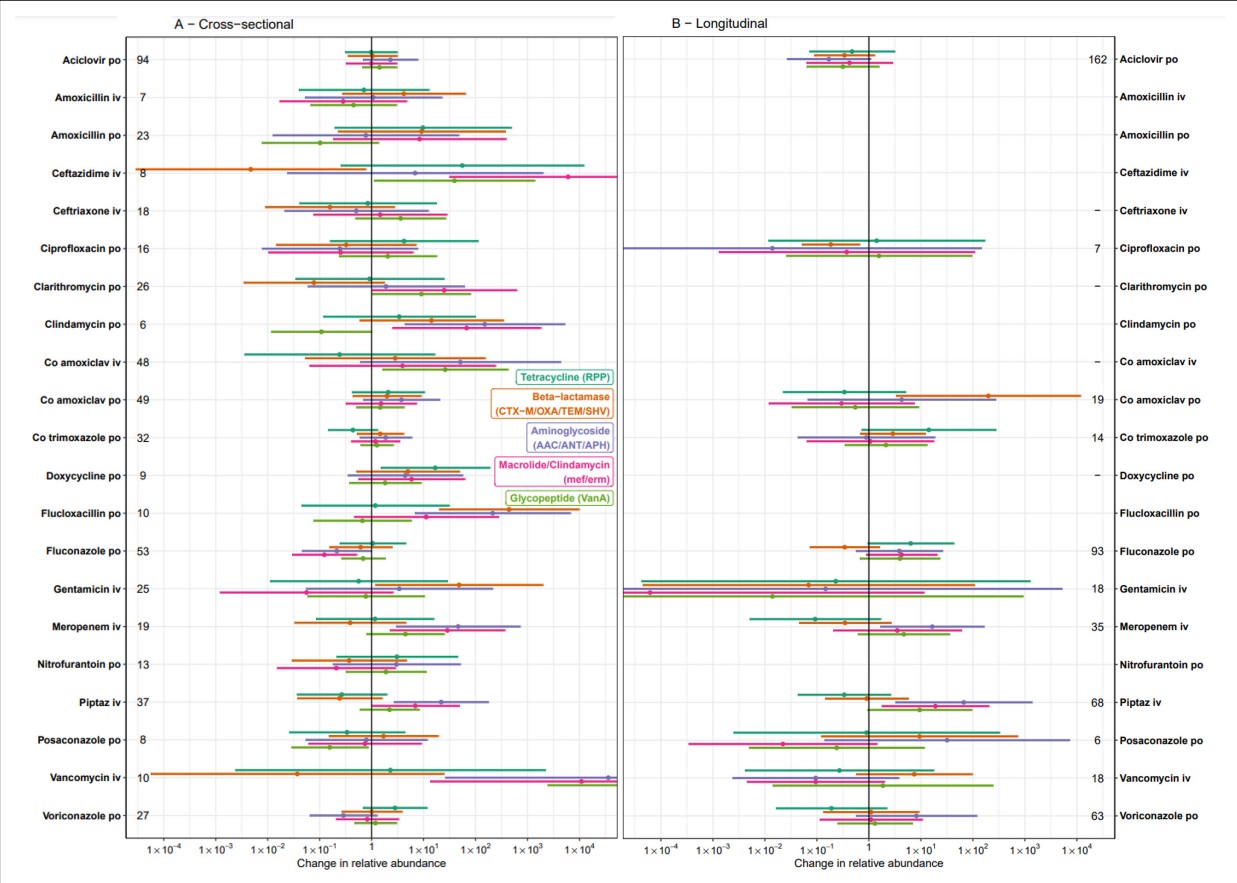

**Figure 3.** Independent effects of specific antimicrobial exposures on relative abundance of selected AMR genes, in (A) cross-sectional and (B) longitudinal multivariable analyses. Error bars represent 95% confidence intervals. Numbers by each exposure represent sample size (n). Non-antimicrobial covariates are not shown but were included in the model and can be found in *Source data 1*. Estimates represent the impact of prolonged use, when exposure ≈ 1 (approximately 42 days, see *Appendix 1—figure 2*).

associated with large increases in the relative abundance of *Enterococcus* in both models (>100 fold increased relative abundance, p<0.01), as were iv co-amoxiclav (p=0.04), iv ceftriaxone (p=0.02), and iv vancomycin (p=0.03) in the cross-sectional model. Many antibiotics were associated with large reductions in the relative abundance of major anaerobe groups, particularly Actinobacteria (reductions in which were associated with exposure to ceftazidime, oral ciprofloxacin, oral clindamycin, iv co-amoxiclav, meropenem, piperacillin-tazobactam, and iv vancomycin). When analysed by antimicrobial class, broad-spectrum beta-lactams were associated with reductions in all anaerobe groups (*Appendix 1—figure 5*).

Several antimicrobial exposures were also associated with increases in the relative abundance of AMR genes (*Figure 3*, *Appendix 1—figure 6*). In the cross-sectional analysis, iv piperacillin-tazobactam and meropenem were both associated with increases in several AMR mechanisms, including aminoglycoside transferases and macrolide resistance genes (all p<0.02), but there was no evidence that these antibiotics had any effect on overall relative abundance of beta-lactamases (p≥0.14), potentially reflecting lower abundance of relevant species (see above) but greater AMR gene carriage in those surviving. In the cross-sectional model, exposure to several classes of antimicrobials was associated with an increase in the abundance of corresponding resistance genes, including glycopeptides, tetracyclines, macrolides and clindamycin (all p≤0.02). Broad-spectrum beta-lactam use was not associated with an increase in the abundance of beta-lactamases, but it was associated with an increase in the abundance of glycopeptide and aminoglycoside AMR genes (both p≤0.0001). Increased aminoglycoside AMR gene abundance was also associated with clindamycin and glycopeptide use (both p≤0.02), but not aminoglycoside use.

## Discussion

Our findings demonstrate that simultaneous modelling of multiple antimicrobial exposures in a heterogeneous and heavily antibiotic-exposed population can produce direct, quantitative comparisons of the impact of different agents on multiple aspects of the gut microbiome. Several broad-spectrum antibiotics were associated with large and rapid reductions in gut microbiome diversity, including decreased abundance of several major anaerobe groups and increased abundance of *Enterococcus* species, along with glycopeptide and aminoglycoside resistance mechanisms often found in *Enterococcus faecium* (*Miller et al., 2014*). The plausibility of these results is reinforced by the consistency between the two independent analysis frameworks that were used: cross-sectional and longitudinal. Microbiome disruption was clearest with clindamycin and broad-spectrum beta-lactams including iv co-amoxiclav, piperacillin-tazobactam, and meropenem, consistent with some previous microbiome studies of these drugs (*Rashid et al., 2015*; *Kabbani et al., 2017*; *MacPherson et al., 2018*; *Shono et al., 2016*). This is in keeping with estimates of risk of *Clostridioides difficile* infection (CDI), which is highest with clindamycin, fluoroquinolones, carbapenems, and third-generation cephalosporins (*Slimings and Riley, 2014*; *Deshpande et al., 2013*). The limited impact of doxycycline on diversity is also consistent with the lower risk of CDI observed with tetracyclines. The decreased relative abundance of *Enterobacteriaceae* seen in this study with piperacillin-tazobactam, ceftriaxone, meropenem, and ciprofloxacin corresponds with reductions seen in culture-based studies (*Sullivan et al., 2001*). The absence of any clear effect of beta-lactams on beta-lactamase gene abundance may be because there were too few patients in this study to create robust categories of beta-lactamase by resistance spectrum, so opposite impacts on different beta-lactamase genes would have been combined.

Observational studies of the gut microbiome allow large numbers of patients to be recruited much more easily than interventional studies such as clinical trials. However, using data from these to inform antibiotic usage is complicated by potential biases from confounding and because of difficulties in accurately modelling multiple exposures in patients receiving many different antibiotics. In ARMORD, the apparent impact of vancomycin and clarithromycin on diversity was substantially reduced when adjusting for other antibiotic exposures (*Figure 1A*), in keeping with the fact that these drugs are often given alongside broad-spectrum beta-lactams in the UK. Nevertheless, the complexity of antibiotic exposure captured in observational data more closely reflects real life, whereas clinical trials generally manipulate one single antibiotic and restrict background antibiotic exposure to produce a cleaner, but potentially less generalisable, comparison.

The large degree of inter-individual variation in the gut microbiome introduces an additional source of variation to cross-sectional studies compared to longitudinal sampling. Despite this, cross-sectional sampling in ARMORD generally gave more precise estimates than longitudinal sampling, despite using a similar number of samples. Recruiting any hospitalised patients for longitudinal sampling before they have received antibiotics is challenging, and this was true in ARMORD, even among patients admitted electively for HCT. In ARMORD, the quasi-experimental approach of longitudinal sampling starting before antimicrobial exposure had no advantage over cross-sectional sampling and complicated recruitment.

The ARMORD study has important limitations. Short-read sequencing was used, meaning the genetic context of AMR genes is uncertain, for example if they are associated with mobile genetic elements or present in opportunistic pathogens (*Boolchandani et al., 2019*). Along with age and sex, several markers of acute illness and comorbidity were included in the model to adjust for potential confounding, but there may be residual and unmeasured confounding related to factors not adequately represented in the model. This could explain some inconsistencies in the effect of specific exposures between the cross-sectional and longitudinal estimates, such as the impact of vancomycin on microbiome composition. The microbiome disruption half-life used was 6 days, as this value best fitted the overall data, which implies that the majority of the disruption and recovery of the bowel flora diversity occurs rapidly after starting and stopping antimicrobials (*Appendix 1—figure 2*). This is in keeping with interventional studies in volunteers (*MacPherson et al., 2018*; *Burdet et al., 2019*), but is a simplification that does not account for some longer-term impacts of antibiotic use that can be detected months after treatment, in particular the presence or absence of individual species or AMR genes (*Dethlefsen et al., 2008*; *Jakobsson et al., 2010*; *Anthony et al., 2022*). A larger study would allow more complex exposure models to be explored, as well as more granular analyses at the level of individual species or resistance genes. Also, because the study was observational, no

data were available on drugs that are not commonly used at OUH, including imipenem, cefepime, and aztreonam, and few data were available for some other drugs, such as clindamycin, limiting the reliability of these estimates. The study focused on participants with healthcare exposure, as it is this setting where antimicrobials are most likely to be used and where microbiome disruption or AMR selection is of most consequence. However, this limited its ability to assess the impact of some potentially relevant exposures that were uncommon in our population, such as recent foreign travel, which is associated with the acquisition of AMR genes (*Worby et al., 2023*). Finally, although we estimate the independent effects of multiple exposures, the uncertainty is still large, making it challenging to use these results to adequately inform clinical use, as this would require clear distinction between antibiotics that might be given for the same indication. For example, the cross-sectional data are consistent with an identical average reduction in diversity with piperacillin-tazobactam, meropenem, ceftazidime, ceftriaxone, intravenous co-amoxiclav, ciprofloxacin and clindamycin, but they are also consistent with important differences between these drugs. A larger study is required to identify or rule out such differences.

Overall, however, simultaneous estimation of the impact of over 20 antimicrobials on the gut microbiome and AMR gene abundance highlighted important differences between individual drugs, and that some drugs in the WHO Access group (co-amoxiclav, clindamycin) had similar magnitude impact on microbiome diversity to those in the Watch group (meropenem, piperacillin-tazobactam) with potential implications for acquisition of other resistant organisms. The consistency of the ARMORD results between sampling frames and with previous studies supports the wider use of observational metagenomic studies to compare the impact of antimicrobials on the gut microbiome. Although some challenges remain, such as identifying an optimal measure of antimicrobial exposure, this is a practical approach to inform future research and stewardship.

## Acknowledgements

We thank all the participants who took part in ARMORD, and also the nurses, healthcare support workers, and doctors on the OUH Haematology ward for their help with this study. This work was supported by the National Institute for Health and Care Research Health Protection Research Unit (NIHR HPRU) in Healthcare Associated Infections and Antimicrobial Resistance at Oxford University in partnership with the UK Health Security Agency (NIHR200915), and the NIHR Biomedical Research Centre, Oxford. The views expressed are those of the authors and not necessarily those of the NHS, the NIHR, the Department of Health or the UK Health Security Agency. The funders had no role in study design, data collection and analysis, decision to publish, or preparation of the manuscript.

## Additional information

### Funding

| Funder | Grant reference number | Author |
|---|---|---|
| National Institute for Health and Care Research | NIHR200915 | Ann Sarah Walker |

The funders had no role in study design, data collection and interpretation, or the decision to submit the work for publication.

### Author contributions

Leon Peto, Conceptualization, Data curation, Formal analysis, Investigation, Visualization, Methodology, Writing – original draft, Project administration; Nicola Fawcett, Conceptualization, Data curation, Investigation, Methodology, Project administration, Writing – review and editing; Musaiwale M Kamfose, Claire Scarborough, Investigation; Andy Peniket, Robert Danby, Resources, Writing – review and editing; Timothy EA Peto, Derrick W Crook, Martin J Llewelyn, Ann Sarah Walker, Conceptualization, Resources, Supervision, Writing – review and editing

### Author ORCIDs

Leon Peto (ORCID) https://orcid.org/0000-0002-4100-2163

Derrick W Crook (ID) https://orcid.org/0000-0002-0590-2850
Ann Sarah Walker (ID) https://orcid.org/0000-0002-0412-8509

## Ethics

The study was approved by the East Midlands-Leicester Central Research Ethics Committee (15/EM/0270). Participants were eligible if they were ≥18 years old, had no stoma or active inflammatory bowel disease, and were able to give informed consent and provide a history of recent antibiotic use.

Reviewer #2 (Public review): https://doi.org/10.7554/eLife.97751.3.sa1
Author response https://doi.org/10.7554/eLife.97751.3.sa2

---

## Additional files

### Supplementary files

Supplementary file 1. ARMORD case report form.

MDAR checklist

Source code 1. R project allowing reproduction of the analyses in this paper.

Source data 1. Deidentified study data and outputs used to plot *Figures 1–3* and *Appendix 1—figures 4–6*.

### Data availability

The data and code used to produce this manuscript are available in the supplementary material, including processed microbiome data, and pseudonymised patient metadata. The sequence data for this study have been deposited in the European Nucleotide Archive (ENA) at EMBL-EBI under accession number PRJEB86785.

The following dataset was generated:

| Author(s) | Year | Dataset title | Dataset URL | Database and Identifier |
| --- | --- | --- | --- | --- |
| Peto L, Fawcett N | 2025 | The Antibiotic Resistance in the Microbiome - Oxford | https://www.ebi.ac.uk/ena/browser/view/PRJEB86785 | European Nucleotide Archive, PRJEB86785 |

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

# Appendix 1

## Supplementary methods

### Laboratory methods

DNA extraction was performed using bead beating followed by QIAGEN Fast DNA Stool Minikit using the following protocol:

### Preparation

1) Prepare 2 ml Lysing Matrix E tube (MP Biomedicals) by adding 1 ml Stool Transport and Recovery buffer (Roche) and *T. thermophilus* DNA (DSMZ, typically 100 ng, but amount varied between extractions). Label and weigh tubes.

2) Remove stool samples from –80 °C to thaw immediately before extraction.

### Lysis

3) Add ~20–1000 mg sample to each tube, transferring more from more liquid samples. Weigh again to establish the weight of the sample.

4) Bead beat tubes twice at 6 m/s for 40 s with a FastPrep-24 5 G instrument (MPBiomedicals) with 5 min interval at 4 °C

5) Incubate tubes at 95 °C for 5 min and centrifuge at 1000 × $g$ for 1 min

6) Transfer 900 µl supernatant to 1 ml InhibitEX buffer in 2 ml Eppendorf, invert x 100 and centrifuge at 17,000 × $g$ for 3 min

7) Transfer 400 µl supernatant to 30 µl proteinase K in 1.5 ml Eppendorf

8) Add 400 µl AL lysis buffer and invert x 100

9) Incubate tubes at 70 °C for 1 min and centrifuge at 17,000 × $g$ for 30 s

10) Add 400 µl 100% ethanol, invert x 20 and centrifuge at 17,000 × $g$ for 30 s

### DNA binding and washing

11) Transfer 1st half of supernatant to spin column, centrifuge at 17,000 × $g$ for 1 min and transfer column to new collection tube

12) Transfer 2nd half of supernatant to same spin column, centrifuge at 17,000 × $g$ for 1 min and transfer column to new collection tube

13) Add 500 µl AW1 wash buffer to column, centrifuge at 17,000 × $g$ for 1 min and transfer column to new collection tube

14) Add 500 µl AW2 wash buffer to column, centrifuge at 17,000 × $g$ for 1 min and transfer column to new collection tube

15) Centrifuge at 17,000 × $g$ for 1 min to dry and transfer column to 1.5 ml Eppendorf Elution

16) Add 50 µl molecular water and incubate at 55 °C for 10 minutes before centrifuging at 17,000 × $g$ for 1 min to elute DNA

Following extraction, DNA concentration was measured with Picogreen and Qubit fluorimeters. DNA was transferred to 96-well plates and stored at –20 °C until sequencing.

Laboratory protocols were designed to minimise the potential for DNA contamination, for example only opening samples in a cabinet, minimising aerosol generation, and never having two sample tubes open simultaneously. Two negative controls were included in each extraction and sequencing run. Samples and negative controls were spiked with a fixed mass of DNA from *Thermus thermophilus* (an extremophile bacterium not found in human flora) to allow quantification of extraneous DNA contamination. The median relative abundance of contaminating DNA in sequenced samples was estimated to be $6.1 \times 10^{-5}$, which was considered to have a negligible impact on microbiome measures.

## Bioinformatic methods

SAMtools (v1.7) was used to reformat and merge sequence data, and BBDuk (v37.90) was used to remove Illumina adapters and quality-trim reads using a phred-score cut-off of 10. Human DNA

reads were identified using the Kraken2 taxonomic classifier and removed prior to analysis (v2.0.6). Quality control metrics were assessed with FastQC (v0.11.7) and MultiQC (v1.5). All samples were subsampled to a depth of 3.5 million paired reads using BBDuk, and samples with fewer reads were not used in this analysis. Taxonomic classification was also performed with MetaPhlAn2 (v2.9.20 using database v292) using a standard database containing Bacteria, Archaea and viruses from RefSeq, plus the human genome. This was used to calculate the Shannon diversity index for each sample (the sum of -p*ln(p) for all species, where p is proportional abundance). Kraken2 was used to assess the abundance of specific bacterial taxa. AMR gene detection in metagenomic sequence data was performed with the ARIBA software package (v2.11.1), using the CARD database and ontology (v3.0.2). Analysis was performed in R v4.2.3 with the ontologyIndex package (v2.4).

## Outcomes

The following bacterial taxa of interest identified using Kraken2:

1. Enterobacteriaceae (NCBI taxonomy ID 543)
2. Enterococcus (ID 1350)
3. Bacteroidetes (ID 976)
4. Clostridia (ID 186801)
5. Actinobacteria (ID 201174)

The following AMR gene classes of interest were identified using ARIBA using the Comprehensive Antibiotic Resistance Database (CARD):

1. Beta-lactamases (TEM [CARD accession ARO:3000014], SHV [ARO:3000015], CTX-M [ARO:3000016], and OXA [ARO:3000017])
2. Tetracycline-resistant ribosomal protection proteins (ARO:0000002)
3. Aminoglycoside transferases (AACs [ARO:3000121], ANTs [ARO:3000218], and APHs [ARO:3000114])
4. Macrolide resistance genes (Erm 23 S ribosomal RNA methyltransferases [ARO:3000560], and Mef efflux pumps [ARO:3000747])
5. VanA (ARO:3000010)

## Statistical methods

A representation of the antimicrobial exposure model is shown below (***Appendix 1—figure 1***). This hypothetical patient has received four courses of treatment with three different agents prior to sample collection. At the time of sample collection, the patient is being treated with drug B alone. Treatment courses are defined as running from the time of the initial dose until the final dose plus one dosing interval (which ranges from 6 to 24 hr depending on the agent). The sum of the area under the curve for each agent is used to calculate the total exposure to that agent:

$$\int_0^x 2^{-x/\lambda} dx = \frac{1 - 2^{-x/\lambda}}{ln2}$$

In which $\lambda$ is the microbiome disruption half-life, and x is time before sample collection. The total area under the curve is defined as 1 (i.e. exposures above are multiplied by *ln*2), so exposure to each antimicrobial is in the range 0–1. Exposures to the same antibiotic at separate times are added to provide the total exposure (so in this example, exposure to antimicrobial B is the sum of two areas shaded red). Exposures to different antimicrobials are independent of one another, as are exposures to the same antimicrobial via different routes of administration.

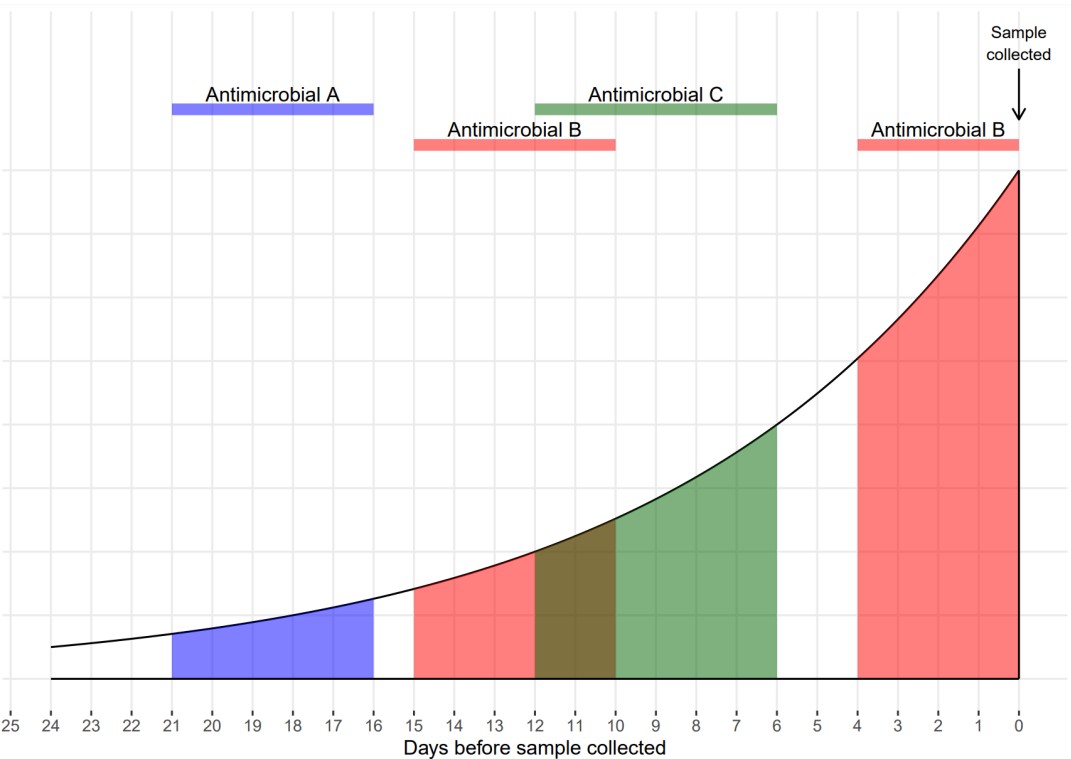

**Appendix 1—figure 1.** Depiction of antimicrobial exposure model. Modelled exposure for each antimicrobial is equal to the sum of shaded areas. The microbiome disruption half-life (lambda) in this example is 6 days.

A single value of $\lambda$ was used for all analyses, chosen as the common value across all antimicrobial exposures with the lowest Akaike Information criterion across 1–14 days in the cross-sectional model with an outcome of Shannon diversity (this model was used to derive $\lambda$ because the cross-sectional model had the greatest amount of exposure data, and Shannon diversity was calculable without imputation for all samples). The optimal value of $\lambda$ was 6 days (*Appendix 1—table 1*), which was used for all subsequent analyses. The exposure for samples taken at various time points during or after a course of antimicrobials is shown in *Appendix 1—figure 2*.

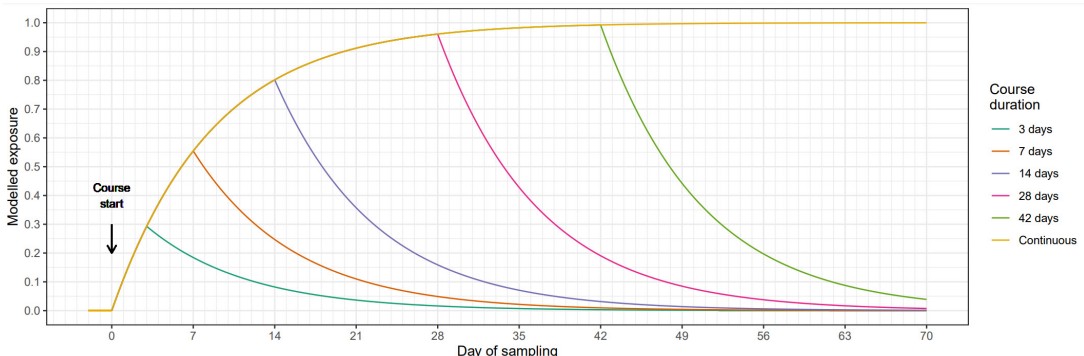

**Appendix 1—figure 2.** Modelled exposure to antimicrobial courses of varying duration. Modelled exposure to a single course of varying duration, starting at day 0. The microbiome disruption half-life (lambda) in this example is 6 days.

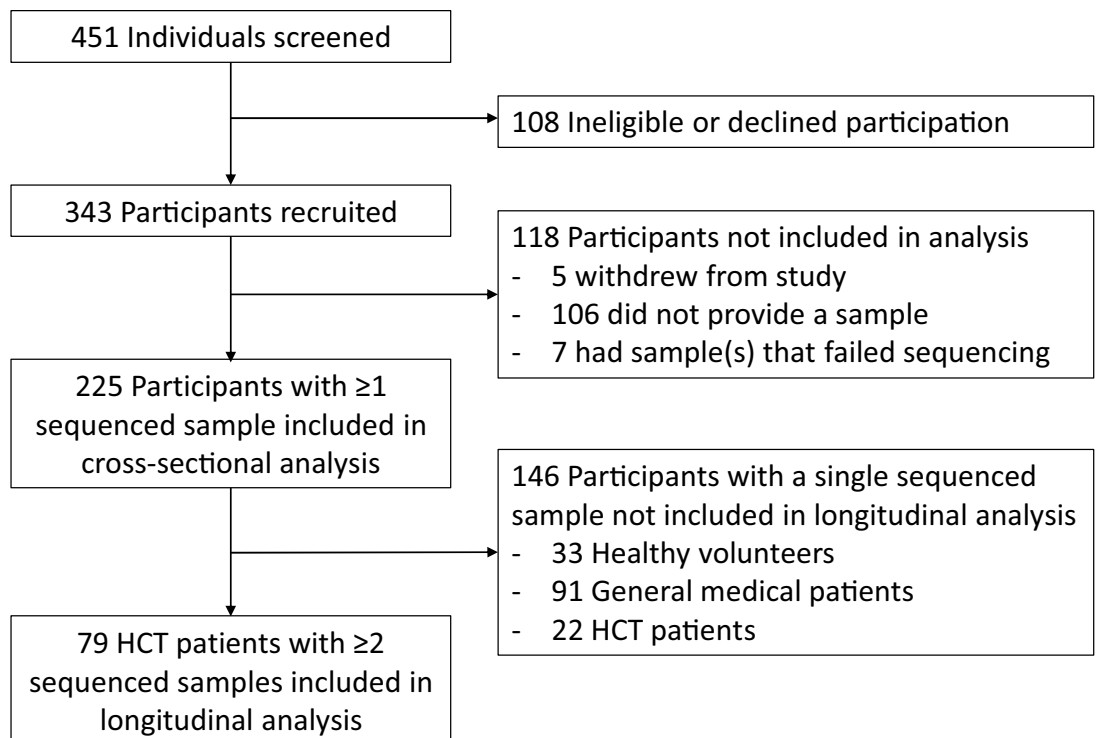

**Appendix 1—figure 3.** ARMORD study CONSORT diagram. HCT = Haematopoietic cell transplant.

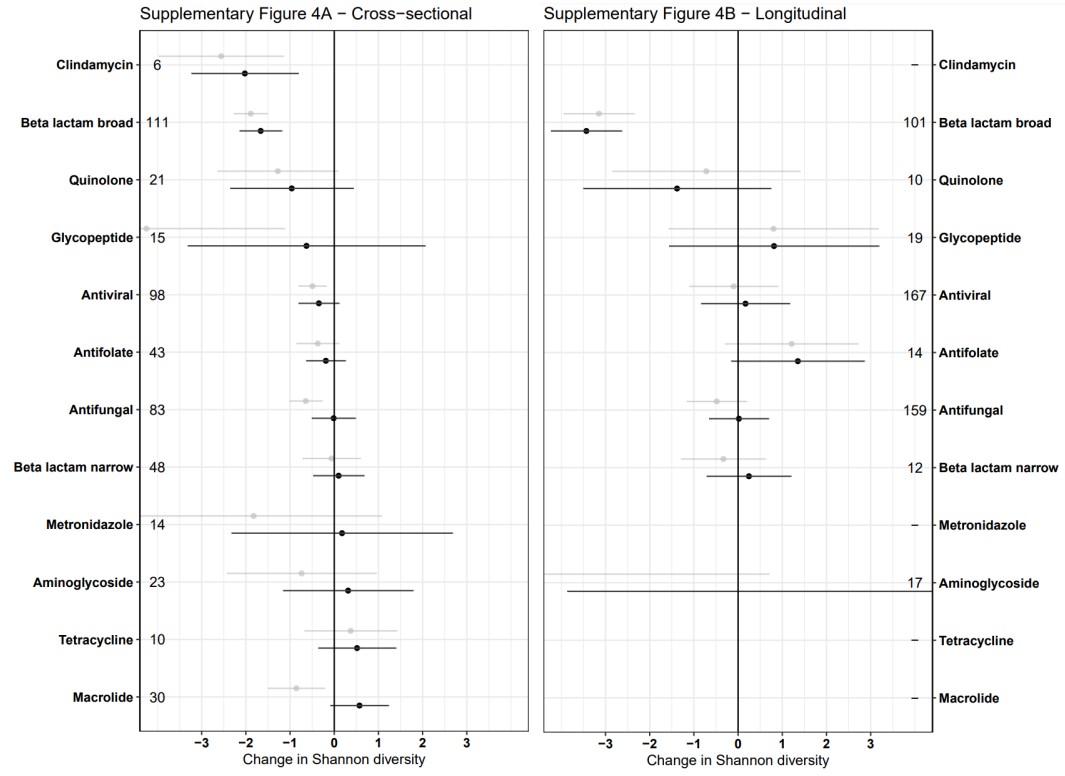

**Appendix 1—figure 4.** Independent effects of exposure to different antimicrobial classes on Shannon diversity in (A) cross-sectional and (B) longitudinal analysis. Multivariable estimates are in black, univariable (unadjusted) estimates in grey. Error bars represent 95% confidence intervals. Numbers by each exposure represent sample size (n). Non-antimicrobial covariates are not plotted here but were included in the model. 'Narrow' beta-lactams

*Appendix 1—figure 4 continued on next page*

*Appendix 1—figure 4 continued*
are penicillin, amoxicillin, flucloxacillin, and first-generation cephalosporins; all others are defined as 'broad'. Estimates represent the impact of prolonged use, when exposure ≈ 1 (approximately 42 days, see *Appendix 1— figure 2*).

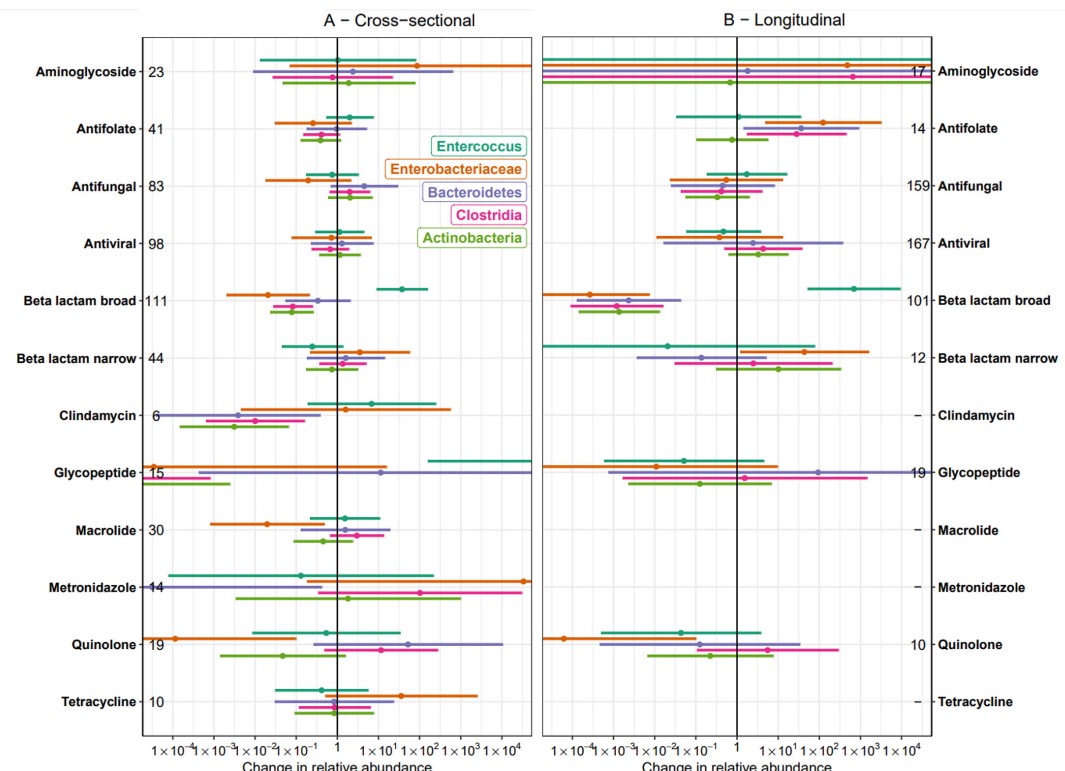

**Appendix 1—figure 5.** Independent effects of exposure to different antimicrobial classes on relative abundance of selected taxa in (A) cross-sectional and (B) longitudinal analysis. Error bars represent 95% confidence intervals. Numbers by each exposure represent sample size (n). Non-antimicrobial covariates are not shown but were included in the model. Antimicrobial categories are the same as *Appendix 1—figure 4*. Estimates represent the impact of prolonged use, when exposure ≈ 1 (approximately 42 days, see *Appendix 1—figure 2*).

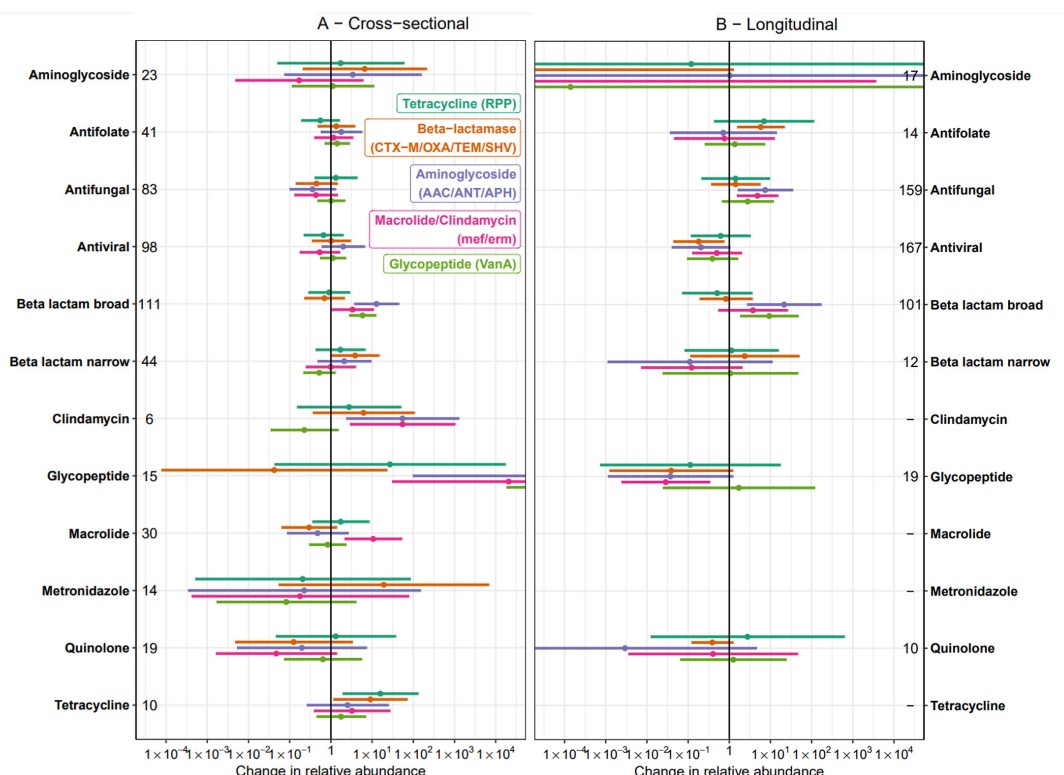

**Appendix 1—figure 6.** Independent effects of exposure to different antimicrobial classes on relative abundance of selected AMR genes in (A) cross-sectional and (B) longitudinal analysis. Error bars represent 95% confidence intervals. Numbers by each exposure represent sample size (n). Non-antimicrobial covariates are not shown but were included in the model. Antimicrobial categories are the same as *Appendix 1—figure 4*. Estimates represent the impact of prolonged use, when exposure ≈ 1 (approximately 42 days, see *Appendix 1—figure 2*).

**Appendix 1—table 1.** Identification of best-fit antimicrobial exposure half-life.
The microbiome disruption half-life with the lowest AIC (i.e. best fit) was used for subsequent analyses.

| Model half-life (days) | $R^2$ | Adjusted $R^2$ | Akaike Information Criterion (AIC) |
|---|---|---|---|
| 1 | 0.4078 | 0.3162 | 477.3 |
| 2 | 0.4331 | 0.3351 | 473.47 |
| 3 | 0.4583 | 0.358 | 467.24 |
| 4 | 0.4676 | 0.369 | 463.35 |
| 5 | 0.4747 | 0.3774 | 460.33 |
| **6** | **0.505** | **0.4102** | **448.95** |
| 7 | 0.5048 | 0.41 | 449.05 |
| 8 | 0.5035 | 0.4084 | 449.65 |
| 9 | 0.5015 | 0.406 | 450.55 |
| 10 | 0.4993 | 0.4035 | 451.51 |
| 14 | 0.4925 | 0.3921 | 456.57 |

**Appendix 1—table 2.** Relative abundance of major taxa in baseline samples.

| Taxon | Median relative abundance (IQR) % |
|---|---|
| *Enterococcus* | 0.13 (0.056–1.3) |

*Appendix 1—table 2 Continued on next page*

Appendix 1—table 2 Continued

| Taxon | Median relative abundance (IQR) % |
| --- | --- |
| *Enterobacteriaceae* | 1.4 (0.055–7.7) |
| Bacteroidetes | 38 (18–61) |
| Clostridia | 22 (10–40) |
| Actinobacteria | 3.3 (0.70–9.1) |
| *All taxa above* | *91 (85–96)* |

