## [Editor Report · eLife Assessment]

This study offers a **valuable** assessment of the impact of antibiotics on the human gut microbiota across diverse observational cohorts. The findings presented are **convincing**, despite the observational design and residual confounding that may still contribute to discrepancies between the cross-sectional and longitudinal data. The work is relevant for researchers and clinicians interested in antimicrobial resistance and the impact of antibiotics on the host.

---

## [Referee Report · Reviewer #2 (Public review)]

Summary:

In this manuscript by Peto et al., the authors describe the impact of different antimicrobials on gut microbiota in a prospective observational study of 225 participants (healthy volunteers, inpatients and outpatients). Both cross-sectional data (all participants) and longitudinal data (subset of 79 haematopoietic cell transplant patients) were used. Using metagenomic sequencing, they estimated the impact of antibiotic exposure on gut microbiota composition and resistance genes. In their models, the authors aim to correct for potential confounders (e.g. demographics, non-antimicrobial exposures and physiological abnormalities), and for differences in the recency and total duration of antibiotic exposure. I consider these comprehensive models an important strength of this observational study. Yet, the underlying assumptions of such models may have impacted the study findings and residual confounding is likely. Other strengths include the presence of both cross-sectional and longitudinal exposure data and presence of both healthy volunteers and patients. Together, these observational findings expand on previous studies (both observational and RCTs) describing the impact of antimicrobials on gut microbiota.

Weaknesses:

(1) The main weaknesses result from the observational design. This hampers causal interpretation and makes correction for potential confounding necessary. While the authors have used comprehensive models to correct for potential confounders and for differences between participants in duration of antibiotic exposure and time between exposure and sample collection, I believe residual confounding is likely (which is mentioned as a limitation in the discussion).

For their models, the authors found a disruption half-life of 6 days to be the best fit based on Shannon diversity. Yet, the disruption caused by antimicrobials may be longer than represented in this model - as highlighted in the discussion.

(2) Another consequence of the observational design of this study is the relatively small number of participants available for some comparisons (e.g. oral clindamycin was only used by 6 participants). Care should be taken when drawing any conclusions from such small numbers.

Comments on revisions:

The authors have adequately addressed all of my comments.

---

## [Author Response]

The following is the authors’ response to the original reviews

**Public Reviews:**

**Reviewer #1 (Public Review):**
Summary:In this manuscript, the authors provide a study among healthy individuals, general medical patients and patients receiving haematopoietic cell transplants (HCT) to study the gut microbiome through shotgun metagenomic sequencing of stool samples. The first two groups were sampled once, while the patients receiving HCT were sampled longitudinally. A range of metadata (including current and previous (up to 1 year before sampling) antibiotic use) was recorded for all sampled individuals. The authors then performed shotgun metagenomic sequencing (using the Illumina platform) and performed bioinformatic analyses on these data to determine the composition and diversity of the gut microbiota and the antibiotic resistance genes therein. The authors conclude, on the basis of these analyses, that some antibiotics had a large impact on gut microbiota diversity, and could select opportunistic pathogens and/or antibiotic resistance genes in the gut microbiota.Strengths:The major strength of this study is the considerable achievement of performing this observational study in a large cohort of individuals. Studies into the impact of antibiotic therapy on the gut microbiota are difficult to organise, perform and interpret, and this work follows state-of-the-art methodologies to achieve its goals. The authors have achieved their objectives and the conclusion they draw on the impact of different antibiotics and their impact on the gut microbiota and its antibiotic resistance genes (the 'resistome', in short), are supported by the data presented in this work.Weaknesses:The weaknesses are the lack of information on the different resistance genes that have been identified and which could have been supplied as Supplementary Data.

We have now supplied a list of individual resistance genes as supplementary data.

In addition, no attempt is made to assess whether the identified resistance genes are associated with mobile genetic elements and/or (opportunistic) pathogens in the gut. While this is challenging with short-read data, alternative approaches like long-read metagenomics, Hi-C and/or culture-based profiling of bacterial communities could have been employed to further strengthen this work.

We agree this is a limitation, and we now refer to this in the discussion. Unfortunately we did not have funding to perform additional profiling of the samples that would have provided more information about the genetic context of the AMR genes identified.

Unfortunately, the authors have not attempted to perform corrections for multiple testing because many antibiotic exposures were correlated.

The reviewer is correct that we did not perform formal correction for multiple testing. This was because correlation between antimicrobial exposures meant we could not determine what correction would be appropriate and not overly conservative. We now describe this more clearly in the statistical analysis section.

Impact:The work may impact policies on the use of antibiotics, as those drugs that have major impacts on the diversity of the gut microbiota and select for antibiotic resistance genes in the gut are better avoided. However, the primary rationale for antibiotic therapy will remain the clinical effectiveness of antimicrobial drugs, and the impact on the gut microbiota and resistome will be secondary to these considerations.

We agree that the primary consideration guiding antimicrobial therapy will usually be clinical effectiveness. However antimicrobial stewardship to minimise microbiome disruption and AMR selection is an increasingly important consideration, particularly as choices can often be made between different antibiotics that are likely to be equally clinically effective.

**Reviewer #2 (Public Review):**
Summary:In this manuscript by Peto et al., the authors describe the impact of different antimicrobials on gut microbiota in a prospective observational study of 225 participants (healthy volunteers, inpatients and outpatients). Both cross-sectional data (all participants) and longitudinal data (a subset of 79 haematopoietic cell transplant patients) were used. Using metagenomic sequencing, they estimated the impact of antibiotic exposure on gut microbiota composition and resistance genes. In their models, the authors aim to correct for potential confounders (e.g. demographics, non-antimicrobial exposures and physiological abnormalities), and for differences in the recency and total duration of antibiotic exposure. I consider these comprehensive models an important strength of this observational study. Yet, the underlying assumptions of such models may have impacted the study findings (detailed below). Other strengths include the presence of both cross-sectional and longitudinal exposure data and the presence of both healthy volunteers and patients. Together, these observational findings expand on previous studies (both observational and RCTs) describing the impact of antimicrobials on gut microbiota.Weaknesses:(1) The main weaknesses result from the observational design. This hampers causal interpretation and corrects for potential confounding necessary. The authors have used comprehensive models to correct for potential confounders and for differences between participants in duration of antibiotic exposure and time between exposure and sample collection. I wonder if some of the choices made by the authors did affect these findings. For example, the authors did not include travel in the final model, but travel (most importantly, south Asia) may result in the acquisition of AMR genes (Worby et al., Lancet Microbe 2023; PMID 37716364). Moreover, non-antimicrobial drugs (such as proton pump inhibitors) were not included but these have a well-known impact on gut microbiota and might be linked with exposure to antimicrobial drugs. Residual confounding may underlie some of the unexplained discrepancies between the cross-sectional and longitudinal data (e.g. for vancomycin).

We agree that the observational design means there is the potential for confounding, which, as the reviewer notes, we attempt to account for as far as possible in the multivariable models presented. We cannot exclude the possibility of residual confounding, and we highlight this as a limitation in the discussion. We have expanded on this limitation, and mention it as a possible explanation for inconsistencies between longitudinal and cross sectional models. Conducting randomised trials to assess the impacts of multiple antimicrobials in sick, hospitalised patients would be exceptionally difficult, and so it is hard to avoid reliance on observational data in these settings.

We did record participants’ foreign travel and diet, but these exposures were not included in our models as they were not independently associated with an impact on the microbiome and their inclusion did not materially affect other estimates. However, because most participants were recruited from a healthcare setting, few had recent foreign travel and so this study was not well powered to assess the effects of travel on AMR carriage. We have added this as a limitation.

In addition, the authors found a disruption half-life of 6 days to be the best fit based on Shannon diversity. If I'm understanding correctly, this results in a near-zero modelled exposure of a 14-day-course after 70 days (purple line; Supplementary Figure 2). However, it has been described that microbiota composition and resistome (not Shannon diversity!) remain altered for longer periods of time after (certain) antibiotic exposures (e.g. Anthony et al., Cell Reports 2022; PMID 35417701). The authors did not assess whether extending the disruption half-life would alter their conclusions.

The reviewer is correct that the best fit disruption half-life of 6 days means the model assumes near-zero exposure by 70 days. We appreciate that antimicrobials can cause longer-term disruption than is represented in our model, and we refer to this in the discussion (we had cited two papers supporting this, and we are grateful for the additional reference above, which we have added). We agree that it is useful to clarify that the longer term effects may be seen in individual components of the microbiome or AMR genes, but not in overall measures of diversity, so have added this to the discussion.

(2) Another consequence of the observational design of this study is the relatively small number of participants available for some comparisons (e.g. oral clindamycin was only used by 6 participants). Care should be taken when drawing any conclusions from such small numbers.

We agree. Although our participants received a large number of different antimicrobial exposures, these were dependent on routine clinical practice at our centre and we lack data on many potentially important exposures. We had mentioned this in relation to antimicrobials not used at our centre, and have now clarified in the discussion that this also limits reliability of estimates for antimicrobials that were rarely used in study participants.

(3) The authors assessed log-transformed relative abundances of specific bacteria after subsampling to 3.5 million reads. While I agree that some kind of data transformation is probably preferable, these methods do not address the compositional data of microbiome data and using a pseudocount (10-6) is necessary for absent (i.e. undetected) taxa [Gloor et al., Front Microbiol 2017; PMID 29187837]. Given the centrality of these relative abundances to their conclusions, a sensitivity analysis using compositionally-aware methods (such as a centred log-ratio (clr) transformation) would have added robustness to their findings.

We agree that using a pseudocount is necessary for undetected taxa, which we have done assuming undetected taxa had an abundance of 10^-6^ (based on the lower limit of detection at the depth we sequenced). We refer to this as truncation in the methods section, but for clarity we have now also described this as a pseudocount. Because our analysis focusses on major taxa that are almost ubiquitous in the human gut microbiome, a pseudocount was only used for 3 samples that had no detectable Enterobacteriaciae.

We are aware that compositionally-aware methods are often used with microbiome data, and for some analyses these are necessary to avoid introducing spurious correlations. However the flaws in non-compositional analyses outlined in Gloor et al do not affect the analyses in this paper:

(1) The problems related to differing sequence depths or inadequate normalisation do not apply to our dataset, as we took a random subset of 3.5 million reads from all samples (Gloor et al correctly point out that this method has the drawback of losing some information, but it avoids problems related to variable sequencing depth)

(2) The remainder Gloor et al critiques multivariate analyses that assess correlations between multiple microbiome measurements made on the same sample, starting with a dissimilarity matrix. With compositional data these can lead to spurious correlations, as measurements on an individual sample are not independent of other measurements made on the same sample. In contrast, our analyses do not use a dissimilarity matrix, but evaluate the association of multiple non-microbiome covariates (e.g. antibiotic exposures, age) with single microbiome measures. We use a separate model for each of 11 specified microbiome components, and display these results side-by side. This does not lead to the same problem of spurious correlation as analyses of dissimilarity matrices. However, it does mean that estimates of effects on each taxa outcome have to be interpreted in the context of estimates on the other taxa. Specifically, in our models, the associations of antimicrobial exposure with different taxa/AMR genes are not necessarily independent of each other (e.g. if an antimicrobial eradicated only one taxon then it would be associated with an increase in others). This is not a spurious correlation, and makes intuitive sense when using relative abundance as outcome. However, we agree this should be made more explicit.

For these reasons, at this stage we would prefer not to increase the complexity of the manuscript by adding a sensitivity analysis.

(4) An overall description of gut microbiota composition and resistome of the included participants is missing. This makes it difficult to compare the current study population to other studies. In addition, for correct interpretation of the findings, it would have been helpful if the reasons for hospital visits of the general medical patients were provided.

We have added a summary of microbiome and resistome composition in the results section and new supplementary table 2, and we also now include microbiome and resistome profiles of all samples in the supplementary data. We also provide some more detail about the types of general medical patients included. We are not able to provide a breakdown of the initial reason for admission as this was not collected.

**Recommendations for the authors:**

**Reviewer #1 (Recommendations For The Authors):**
(1) Provide a supplementary table with information on the abundance of individual genes in the samples.

This supplementary data is now included.

(2) Engage with an expert in statistics to discuss how statistical analyses can be improved.

A experienced biostatistician has been involved in this study since its conception, and was involved in planning the analysis and in the responses to these comments.

(3) Typos and other minor corrections:Methods: it is my understanding that litre should be abbreviated with a lowercase l.

Different journals have different house styles: we are happy to follow Editorial guidance.

p. 9: abuindance should be corrected to abundance.

Corrected

p. 9: relative species should be relevant species?

Yes, corrected. Thank you.

p. 9 - 10: can the apparent lack of effect of beta-lactams on beta-lactamase gene abundance be explained by the focus on a small number of beta-lactamase resistance genes that are found in Enterobacteriaceae and which are not particularly prevalent, while other classes of resistance genes (e.g. Bacteroidal beta-lactamases) were excluded?

It is possible that including other beta-lactamases would have led to different results, but as a small number of beta-lactamases in Enterobacteriaceae are of major clinical importance we decided to focus on these (already justified in the Methods). A full list of AMR genes identified is now provided in the supplementary data.

p. 10: beta-lactamse should be beta-lactamase

Corrected

Figure 3A: could the data shown for tetracycline resistance genes be skewed by tetQ, which is probably one of the most abundant resistance genes in the human gut and acts through ribosome protection?

TetQ was included, but only accounted for 23% of reads assigned to tetracycline resistance genes so is unlikely to have skewed the overall result. We limited the analysis to a few major categories of AMR genes and, other than VanA, have avoided presenting results for single genes to limit the degree of multiple testing. We now include the resistome profile for each sample in the supplementary data so that readers can explore the data if desired.

**Reviewer #2 (Recommendations For The Authors):**
(1) Given the importance of obligate anaerobic gut microbiota for human health, it might be interesting to divide antibiotics into categories based on their anti-anaerobic activity and assess whether these antibiotics differ in their effects on gut microbiota.

The large majority of antibiotics used in clinical practice have activity against aerobic bacteria and anaerobic bacteria, so it is not possible to easily categorise them this way. There are two main exceptions (metronidazole and aminoglycosides) but there was insufficient use of these drugs to clearly detect or rule out a difference between them, even when categorising antimicrobials by class, so we prefer not to frame the results in these terms. Also see our comments on this categorisation below.

(2) For estimating the abundance of anaerobic bacteria, three major groups were assessed: Bacteroidetes, Actinobacteria and Clostridia. To me, this seems a bit aspecific. For example, the phylum Bacteroidetes contains some aerobic bacteria (e.g. Flavobacteriia). Would it be possible to provide a more accurate estimation of anaerobic bacteria?

We think that an emphasis on a binary aerobic/anaerobic classification is less biologically meaningful that the more granular genetic classification we use, and its use largely reflects the previous reliance on culture-based methods for bacterial identification. Although some important opportunistic human pathogens are aerobic, it is not clear that the benefit or harm of most gut commensals relates to their oxygen tolerance, and all luminal bacteria exist in an anaerobic environment. As such we prefer not to perform an additional analysis using this category. We are also not sure that this could be done reliably, as many of the taxa are characterised poorly, or not at all.

We appreciate that Bacteroidetes, Actinobacteria and Clostridia are diverse taxa that include many different species, so may seem non-specific, but these were chosen because:

i) they are non-overlapping with Enterobacteriaceae and Enterococcus, the major opportunistic pathogens of clinical relevance, so could be used in parallel, and

ii) they make up the large majority of the gut microbiome in most people and most species are of low pathogenicity, so it is plausible that their disruption might drive colonisation with more pathogenic organisms (or those carrying important AMR genes).

We have more clearly stated this rationale.

(3) A statement on the availability of data and code for analysis is missing. I would highly recommend public sharing of raw sequence data and R code for analysis. If possible, it would be very valuable if processed microbiome data and patient metadata could be shared.

We agree, and these have been submitted as supplementary data. We have added the following statement “The data and code used to produce this manuscript are available in the supplementary material, including processed microbiome data, and pseudonymised patient metadata. The sequence data for this study have been deposited in the European Nucleotide Archive (ENA) at EMBL-EBI under accession number PRJEB86785.”